

**Title: Straw return with diverse nitrogen fertilizer application rates modulate ecosystem services**
**and microbial traits in a meadow soil**
Author names: Yan Duan[1,2], Minghui Cao[1,2,3], Wenling Zhong[1,2,3], Yuming Wang[1,2], Zheng Ni[1,2,3],
Mengxia Zhang[1,2,4], Jiangye Li[5], Yumei Li[6], Xianghai Meng[7], Lifang Wu[1,2,3,*]
[1] The centre for Ion Beam Bioengineering Green Agriculture, Hefei Institutes of Physical Science,
Chinese Academy of Sciences, Hefei 230031, Anhui, China
[2] Zhongke Taihe Experimental Station, Taihe 236626, Anhui, China
[3] School of Life Science, University of Science and Technology of China, Hefei 230027, Anhui, China
[4] School of Life Sciences, Anhui Agricultural University, Hefei 230036, China
[5] Institute of Agricultural Resources and Environment, Jiangsu Academy of Agricultural Sciences,
Nanjing 210014, China
[6] Institute of Soil Fertilizer and Environment Resources, Heilongjiang Academy of Agricultural
Sciences, Harbin 150086, China
[7] Mudanjiang Branch of Heilongjiang Academy of Agricultural Sciences, Mudanjiang, 157400, China
[*] Corresponding author: Prof. Lifang Wu. The centre for Ion Beam Bioengineering Green Agriculture,
Hefei Institutes of Physical Science, Chinese Academy of Sciences, Hefei 230031, Anhui, China
E-mail: lfwu@ipp.ac.cn, Tel: +86-551-6559-1413, Fax: +86-551-6559-2420.
Number of text pages: 43; Number of figures: 6; Number of tables:1



**Abstract:**
Nitrogen (N) fertilization has received worldwide attention due to its benefits to soil fertility and
productivity, but excess N application also causes an array of ecosystem dis-services, such as
greenhouse gas emissions. Generally, soil microorganisms are considered to be involved in upholding a
variety of ecosystem services and dis-services. However, the linkages between soil ecosystem services
and microbial traits under different N fertilizer application rates remain uncertain. To address this, a
4-year in situ field experiment was conducted in a meadow soil on the Northeast China Plain after
straw return with the following treatments combined with regular phosphorus (P) and potassium (K)
fertilization: (i) regular N fertilizer (N+PK); (ii) 25% N fertilizer reduction (0.75N+PK); (iii) 50% N
fertilizer reduction (0.5N+PK); and (IV) no N fertilizer (PK). Ecosystem services, dis-services and
microbial traits responded distinctly to the different N fertilizer rates. Treatment 0.75N+PK had overall
positive effects on soil fertility, productivity, straw decomposition, and microbial abundance and
function and alleviated greenhouse effects due to N deficiency. Meanwhile, 0.75N+PK upregulated
aboveground biomass and soil C:N and thus increased the abundance of genes encoding
cellulose-degrading enzymes, which may imply the potential ability of C and N turnover. In addition,
most observed changes in ecosystem services and dis-services were strongly associated with microbial
modules and keystone taxa. Specifically, the *Lasiosphaeriaceae*-driven module 1 community promoted
straw degradation and C and N release, while the *Terrimonas*-driven module 3 community contributed
to production improvement, which was conducive to soil multifunctionality. Therefore, our results
suggest that straw return with 25% chemical N fertilizer reduction is optimal for achieving ecosystem
services. This study highlights the importance of abiotic and biotic factors in soil health and supports
green agricultural development by optimizing N fertilizer rates in meadow soil after straw return.





**Keywords:** Ecosystem services; Straw return; Nitrogen fertilization; Microbial community; Crop yield


**1. Introduction**

Multiple soil ecosystem services are indicators of soil health (Kihara et al., 2020; Lehmann et al., 2021). Soil ecosystem services refer to the ability of soil to function as a vital living system to sustainably increase crop productivity, improve environmental quality, tackle climate change and promote plant and animal health (de Bello et al., 2010; Tang et al., 2019). In recent decades, anthropogenic activity, such as intensive agriculture, has posed a wide range of threats to agroecosystem services (Robertson et al., 2014; Allen et al., 2015). Irrational or excessive application of chemical fertilizers, especially nitrogen (N), is ubiquitous to achieve high crop yields in response to population surges globally (Shi et al., 2019). In fact, N is considered the essential macronutrient for all biota, while excessive N fertilizer inputs not only reduce soil fertility and productivity but also lead to environmental burdens (Trost et al., 2016). For example, previous studies emphasized that N fertilizer abuse may accelerate greenhouse gas emissions (Huang et al., 2006; Wu et al., 2015) and degrade groundwater quality (Rhymes et al., 2016). Therefore, how to achieve agroecosystem services by regulating N fertilizer application rates is a critical issue that needs to be fully assessed.

Straw return has also been widely applied as a major measure to moderate soil ecosystem services (Xu et al., 2021). Plant residues, as natural organic bioenergy resources, contain abundant N that further affects soil fertility and productivity (Pan et al., 2009; Liu et al., 2014). Thus, the straw-derived N released during degradation is an important source that may serve as a partial substitute for chemical N fertilizer application (Wang et al., 2017; Latifmanesh et al., 2020). However, crop fields suffering





from abundant organic materials usually have low reutilization efficiency (Hou et al., 2020). Generally,
the majority of N in straw is released into the atmosphere as oxynitride, such as nitrous oxide ($N_2O$),
resulting in lower soil organic matter (SOM) formation efficiency (Wang et al., 2019; Sun et al., 2021).
Subsequent literature highlighted that straw return significantly elevates greenhouse gas emissions so
that less than 15% of straw-derived N can be transformed into soil and become SOM (Yin et al., 2018;
Wu et al., 2019). However, the potential for the partial substitution of straw for chemical N fertilizer
application is still unclear. Revealing the mechanisms of efficient straw utilization under diverse N
fertilizer input rates is essential to achieving ecosystem multifunctionality.

Compared with plants and animals, soil contains more microorganisms living in an opaque

environment, making the evaluation of soil ecosystem services more complex (Handa et al., 2014;
Wagg et al., 2014). Agronomic management for such "multifunctionality" has prompted research into
the role that microbes play in providing desired rates of multiple ecosystem processes (Gong et al.,
2020). To our knowledge, fertilization-induced changes in microbial communities and functions are
fundamental to the regulation of a variety of ecosystem multifunctionalities, including SOM formation,
greenhouse gas emissions, litter decomposition, and crop production (Dominati et al., 2014). To date,
we still lack empirical evidence of the linkages among N fertilizers, specific microbial communities or
functions and multiple ecosystem services, and the diverse cropland services driven by complex
microbial traits under different N fertilizer rates are seldom clarified.

Microorganisms contribute to ecosystem services by modulating microbial function, community

composition and succession, which makes understanding the consequences of the changes in microbial
traits crucial for determining different N fertilizer input levels (Bradford et al., 2014). The role of
microorganisms in ecosystem functioning is unequivocal, and these organisms can be recognized as the



key drivers of ecosystem services (Chen et al., 2019a). Generally, bacteria and fungi are the main
drivers of straw labile and recalcitrant component decomposition, respectively (Frey et al., 2013; Ge et
al., 2017). Therefore, the ratio of fungi to bacteria is always considered an indicator during straw
degradation periods (Hogberg et al., 2007). Specifically, the expression levels of the *cbhI* and *GH48*
genes were identified as biomarkers of cellulolytic fungi and bacteria, respectively (Zhang et al., 2017).
Previous studies revealed that the N input level dominated the associations between microbial
composition and cellulolytic gene abundance with SOM physical fractions (Duan et al., 2021). In
addition, microbial module communities and keystone taxa have been used to provide satisfactory
explanations for ecosystem services. Chen et al. (2019b) found that particular microbial modules
participated in N and phosphorus (P) accumulation and $CO_2$ emissions in a Cambisol. Moreover,
specific taxa are involved in agrosystem services. For example, Actinobacteria have been extensively
studied and can be considered the main degraders of straw by secreting cellulase (Bao et al., 2021).
*Mortierella* has been proven to increase soil fertility and crop yield due in part to its strong C
sequestration capacity (Ning et al., 2020). Notably, it is also well known that microbial traits are
mediated by nutrient availability and stoichiometry (Chen et al., 2014). C, N and P stoichiometry has
profound impacts on microbial in vivo metabolism and ex vivo modification processes (Chen et al.,
2016). Multiple studies have indicated that soil C:N and N:P ratios are the key factors mediating
microbial functions and soil health (Ning et al., 2020; Duan et al., 2021). Nevertheless, the knowledge
of the microbial mechanisms that modulate ecosystem services in response to N fertilizer input levels
are still rudimentary.

As an important grain-producing region, the Northeast China Plain contributes to more than 20%

of the total grain yield in China (Li et al., 2017). However, excessive chemical N fertilizer inputs have



caused ecosystem dis-services over the past decades (Zhao et al., 2018). Therefore, a field experiment
was conducted to reveal the influences of N input levels on soil ecosystem multifunctionality and
associated microbial traits and to try to establish the linkages between them. In the present study, two
hypotheses were tested: (i) soil ecosystem services and dis-services would show distinct responses to N
fertilizer input levels, and (ii) the changes in cropland ecosystem services and dis-services would be
linked to specific microbial traits.

**2.  Materials and methods**
**2.1 Site description and sampling**
A field experiment under contrasting inorganic N fertilizer input levels was established in 2018 in
Wenchun town (44°59′61″ N, 129°59′18″ E), Mudanjiang city, Heilongjiang Province, Northeast China
Plain, which is an important grain-producing area. This region has a typical temperate continental
monsoon climate with an average annual temperature of 4.3 °C and a mean annual precipitation of
579.7 mm. The soil is classified as a meadow soil according to US Soil Taxonomy (USST). The
cropping system was continuous maize (*Zea mays* L.) monoculture. Four treatments received different
N fertilizer input levels after straw return to the field for 4 years as follows: (1) regular chemical
fertilization, N+PK (300 kg urea (N 46%) ha$^{-1}$ yr$^{-1}$, 250 kg diammonium phosphate (P$_2$O$_5$ 48%) ha$^{-1}$ yr$^{-1}$,
150 kg potassium chloride (K$_2$O 50%) ha$^{-1}$ yr$^{-1}$); 25% reduction of N fertilizer, 0.75N+PK (225 kg urea
ha$^{-1}$ yr$^{-1}$, 250 kg diammonium phosphate ha$^{-1}$ yr$^{-1}$, 150 kg potassium chloride ha$^{-1}$ yr$^{-1}$); 50% reduction
of N fertilizer, 0.50N+PK (150 kg urea ha$^{-1}$ yr$^{-1}$, 250 kg diammonium phosphate ha$^{-1}$ yr$^{-1}$, 150 kg
potassium chloride ha$^{-1}$ yr$^{-1}$); and no N fertilizer, PK (250 kg diammonium phosphate ha$^{-1}$ yr$^{-1}$, 150 kg
potassium chloride ha$^{-1}$ yr$^{-1}$). All straw and chemical fertilizers were applied with shallow tillage to 20



cm. Straw was cut into pieces less than 5 cm and input after the harvest in October, while the chemical
fertilizers were applied during ploughing in May of the next year. All other normal management
practices were consistent among treatments during the experiment. Before the experiment, the initial
soil contained 18.74 g kg$^{-1}$ SOC, 1.03 g kg$^{-1}$ total N and 0.54 g kg$^{-1}$ total P with a pH of 7.37 ($H_2O$).
The yield and some of the soil chemical properties under different bulk soil treatments during the
experimental process are shown in Supplemental material Table S1.
Soils were sampled after the maize harvest in October 2021. A randomized complete block design
consisting of 5 treatments with 3 replications was adopted in this study. Each field plot was 4.5 m × 15
m. We took nine soil cores (5 cm diameter) from the top 20 cm of bulk soil in each plot. Each soil
sample consisted of a mixture of subsamples randomly collected at nine different positions in the same
plot. In total, 12 soil samples were collected from 4 treatments with 3 replicates. Soils were sieved
through a 2 mm mesh, the mineral particles and plant roots were carefully removed, and then the soils
were homogenized and stored in an incubator at 4 °C in a 40% moisture environment. One part of the
soil sample was air-dried to measure basal soil properties, and the other part was used for microbial
molecular analysis.
**2.2 The field straw decomposition and carbon and nitrogen release experiments**
The ditch-buried straw decomposition experiment was conducted using litter nylon bags. Maize
straw materials were collected after maize harvesting in 2021 and air-dried. Ten grams of maize straw
was cut to 2 cm in length and put into nylon litter bags, which were then sealed via heat sealing. The
nylon bags were 6 cm × 10 cm in size and were made of 200 mesh nylon fabric, which permitted the
free transfer of microorganisms between the nylon bags and soil. Before maize cultivation in 2021,
litter bags containing straw were buried at 10 cm depth in a spatially random design to prevent bags



associated with a given decomposition stage being placed together in space. The litter bags were
collected after the harvest in October 2021.
The straw decomposition ratio was calculated based on dry weight loss as (dry initial mass - dry
final mass)/dry initial mass. The straw-C concentration was measured by titrimetry after oxidation with
a mixture of $H_2SO_4$ and $K_2Cr_2O_7$. Total N, P and K were determined using the Kjeldahl, molybdenum
blue colorimetry, and flame photometry methods, respectively. All methods have been described by Lu
(2000). The initial and sampled maize straw material properties are shown in Supplemental material
Table S2. The amounts of total straw C and N released were calculated by the following equation:
The amounts of total straw C and N released = (initial C (or N) content × dry initial mass - final C
(or N) content × dry final mass) × aboveground biomass
**2.3 Measurement of soil properties and assessment of ecosystem services**
Soil pH was measured at a soil:water ratio of 1:2.5 (weight/weight). Air-dried soil and 25 ml of
deionized water were shaken together for 1 min and left to settle for 30 min, and the soil pH was
determined using an electrode. Soil organic carbon (SOC) was measured by titrimetry after oxidation
with a mixture of $H_2SO_4$ and $K_2Cr_2O_7$. Total N and P were determined using the Kjeldahl and
molybdenum blue colorimetric methods, respectively. All of these methods have been described by Lu

169  (2000).

Microbial biomass C (MBC) and microbial biomass N (MBN) were analysed using the
fumigation-extraction method. Ten grams of fresh soil was fumigated with chloroform in the dark for
24 h, and then the fumigated and nonfumigated soils were extracted with 0.5 M $K_2SO_4$ and shaken at
200 rpm for 0.5 h. Soil extracts were filtered through a 0.45-μm Millipore filter, and the C and N in the
extracts were determined using a multi C/N 3100 analyser (Analytik Jena AG). The C and N contents



in extracts of the nonfumigated soil were subtracted from C and N extracted from the fumigated soil to
give the C and N extracted from the soil microbial biomass. Values of 0.45 and 0.54 were used to
calibrate the contents of MBC and MBN, respectively (Vance et al., 1987; Wu et al., 1990).
The activities of cellulose and N-acetyl-β-glucosaminidase (NAG) were measured using
$p$-nitrophenyl-β-D-cellobioside and $p$-nitrophenyl-N-acetyl-β-D-glucosaminide as substrates,
respectively. Fresh soil (1.0 g) was mixed with 2.5 mL of 0.2 M acetate buffer (pH 5.0) and 2.5 mL of
0.02 M substrates and then shaken at 200 rpm and 37 °C for 1 h. The reaction was stopped by adding 1
mL of 0.5 M $CaCl_2$ and 4 mL of 0.1 M Tris buffer (pH 12.0). The mixture was suspended with a vortex,
the supernatant was filtered, and the concentration of $p$-nitrophenol (PNP) was measured by
colorimetry at 400 nm. The same procedure was followed for the controls, with the exception that the
substrate was added after the incubation, and $CaCl_2$ and Tris buffer were added (Dick, 2011; Geisseler
and Horwath, 2009).
To estimate the greenhouse gas emission potential, we conducted a 60-day incubation experiment.
Briefly, 20 g of fresh soil was placed in a 250-mL flask and then sealed with a gas-tight lid that had a
rubber stopper in the middle. Gas samples (10 mL) were taken from the headspace of each flask at 1, 3,
7, 15, 30, and 60 days after sealing using a plastic syringe. The gas sample was immediately injected
into a preevacuated 10-mL glass vial. Concentrations of methane ($CH_4$), $N_2O$ and carbon dioxide ($CO_2$)
were determined using a gas chromatograph (Agilent 7890) equipped with a flame ionization detector
for $CO_2$ and $CH_4$ and a [63]Ni electron capture detector for $N_2O$. The gas standards were provided by the
National Research Center for Certified Reference Materials, Beijing, China. The precision for
greenhouse gas emission concentrations was ±0.5% based on repeated measurements of gas standards.
When the maize plants matured, all plants and grains were harvested from each plot, oven-dried at



60 °C for 48 h and weighed. Aboveground biomass and crop yield were converted into weight per
hectare.

199        We selected 15 soil properties to estimate cropland ecosystem services, i.e., the soil fertility index

(SOC, total N, total P, MBC and MBN), greenhouse gas emission amount (mainly $CO_2$, $N_2O$ and $CH_4$),
straw decomposition and C and N released, soil extracellular enzymes (cellulase and
N-acetyl-D-glucosaminidase), and maize biomass (aboveground biomass and crop yield). Generally,
SOC, total N and total P are the major soil fertility factors and indicate the present nutrient status in
croplands, which can be used to explain soil fertility conditions. Microbial biomass reflects ecosystem
productivity. Greenhouse gas emissions are related to climate change, which can be regulated by
fertilization regimes and soil microbial activities. Soil extracellular enzymes catalyse the
decomposition of a range of organic polymers, resulting in C and N turnover. Maize biomass (such as
aboveground biomass and crop yield) reflects soil productivity. As a whole, all of these variables
together contributed to the cropland function. To evaluate the function of the cropland ecosystem under
different fertilization conditions, we calculated an integrative soil ecosystem multifunctionality index
for further analysis. Due to the lack of a specific definition of multifunctionality, we first calculated the
*Z* scores of the 15 measured variables and obtained a multifunctionality value for each plot by
averaging the *Z* scores of the 15 variables.
**2.4 DNA extraction and quantification of general fungal ITS, bacterial 16S rRNA and genes**
**encoding cellulose-degrading enzymes**

Total DNA was extracted from 0.5 g freeze-dried soil by using a Fast DNA Spin Kit for Soil

(MPbio, USA) according to the manufacturer's instructions and then dissolved in 50 μl of Tris-EDTA
buffer. The success of the DNA extraction was characterized by electrophoresis on 1% (wt/vol) agarose



gels. The quantity and quality of DNA were checked using a Nanodrop spectrophotometer (Nanodrop,
PeqLab, Germany). The extracted DNA samples were stored at −80 °C before molecular analysis.
Bacterial and fungal abundances were determined to reveal the changes in microbial community
compositions. The abundances of bacteria and bacteria fungi were measured according to modified
procedures (Fierer and Jackson., 2005). We selected the primers *338F/518R (338F:*
*CCTACGGGAGGCAGCAG; 518R: ATTACCGCGGCTGCTGG)* and *NSI1/58A2R (NSI1:*
*GTAGTCATATGCTTGTCT; 58A2R: CATTCCCCGTTACCCGTT)* for the qPCR assay. The thermal
qPCR profiles for the bacteria and fungi were as follows: 95 °C 2 min, 35 cycles (95 °C 30 s, 60 °C 30
s, 72 °C 30 s, 80 °C 15 s), and data collection at 81 °C for 10 s; 95 °C 10 min, 40 cycles (95 °C 15 s,
52 °C 30 s, 72 °C 30 s, 79 °C 30 s), and data collection at 81 °C for 10 s, respectively. The initial
concentrations of the two plasmids used as the standards for the bacterial and fungal abundance
analyses were $1.22 \times 10^{10}$ and $9.05 \times 10^{9}$, respectively.
The fungal *cbhI* gene and bacterial *GH48* gene were selected as functional biomarkers of
cellulolytic fungi and bacteria, respectively. The primers *GH48 F8/GH48 R5 (GH48_F8*: 5 -
GCCADGHTBGGCG ACTACCT - 3; *GH48_R5*: 5 - CGCCCCABGMSWWGTACCA - 3) and *cbhI*
*F/cbhI R (cbhI F*: ACCAAYTGCTAYACIRGYAA; *cbhI R*: GCYTCCCAIATRTCCATC) were used for
the qPCR assay. The abundance of bacterial *GH48* and fungal *cbhI* genes was quantified according to
modified procedures (Zhang et al., 2017). The thermal profiles of qPCR for the target genes of *GH48*
and *cbhI* were as follows: 95 °C for 5 min, 40 × (94 °C for 30 s, 60 °C for 45 s, and 72 °C for 90 s),
and data collection at 84 °C for 10 s; and 94 °C for 4 min, 40 × (94 °C for 45 s, 50 °C for 30 s, and
72 °C for 60 s), and data collection at 81 °C for 10 s, respectively. The initial concentrations of the two
plasmids as the standards for bacterial *GH48* and fungal *cbhI* gene abundance analysis corresponded to



$1.85 \times 10^{11}$ and $2.65 \times 10^{10}$ copies g$^{-1}$ dry soil, respectively. qPCR was performed in triplicate, and
amplification efficiencies higher than 95% were obtained with r$^2$ values > 0.99.
**2.5 Bacterial 16S rRNA genes and fungal ITS amplification and sequencing**

High-throughput sequencing was performed with the Illumina MiSeq sequencing platform

(Illumina Inc.). Both the forward and reverse primers were tagged with an adapter and linker sequence,
and 8-bp barcode oligonucleotides were added to distinguish the amplicons from different soil samples.

The    primers    *515F    (5'-GTGCCAGCMGCCGCGGTAA-3')*    and    *907R*

*(5'-CCGTCAATTCMTTTRAGTTT-3')* were chosen to amplify the 16S rRNA genes in the V4–V5
hypervariable region. PCR was conducted in a 50-µL reaction mixture containing 27 µL of ddH$_2$O, 2
µL (5 µM) of each forward/reverse primer, 2.5 µL (10 ng) of template DNA, 5 µL (2.5 mM) of
deoxynucleoside triphosphates, 10 µL of 5× Fastpfu buffer, 0.5 µL of bovine serum albumin, and 1 µL
of TransStart Fastpfu polymerase (TransGen, Beijing, China). The PCR conditions were 94 °C for 5
min; 30 cycles of 94 °C for 30 s, 52 °C for 30 s and 72 °C for 30 s of extension; followed by 72 °C for
10 min (Caporaso et al., 2010).

The    fungal    ITS1    region    was    amplified    using    the    primer    pair    *ITS1F*

*(CTTGGTCATTTAGAGGAAGTAA)/ITS2 (GCTGCGTTCTTCATCGATGC)* (Ghannoum et al., 2010).
The 50-µL reaction mixture of each reaction mix consisted of 1 µl (30 ng) of DNA, 4 µl (1 µM) of each
forward/reverse primer, 25 µl of PCR Master Mix, and 16 µl of ddH$_2$O. PCR amplification was
conducted at 98 °C for 3 min, followed by 30 cycles (98 °C for 45 s, 55 °C for 45 s, and 72 °C for 45 s),
with a final extension at 72 °C for 7 min (Ghannoum et al., 2010). All amplicons were cleaned and
pooled in equimolar concentrations in a single tube, after which they were subjected to library
preparation, cluster generation, and 250-bp paired-end sequencing on an Illumina MiSeq platform



(Illumina Inc., San Diego, CA, USA).
The raw sequence data were processed using the Qualitative Insights into Microbial Ecology
(QIIME) pipeline (Caporaso et al., 2010). Sequences that fully matched the barcodes were selected and
distributed into separate files for the bacterial 16S rRNA and fungal ITS genes. Poor-quality sequences
with lengths less than 200 bp (for fungal ITS) and 500 bp (for bacterial 16S) and quality scores less
than 20 were discarded, and the chimaeras were removed using the UCHIME algorithm (Edgar et al.,
2010). The remaining sequences were assigned to operational taxonomic units (OTUs) with a 97%
similarity threshold using UCLUST (Edgar, 2010). Alpha diversity and Bray–Curtis distances for
principal coordinate analysis of the soil microbial community were calculated after rarefying all
samples to the same sequencing depth.
**2.6 Statistical analysis**
The soil ecosystem multifunctionality index, crop yields, microbial traits and other relevant soil
variables among treatments were subjected to a chi-square test for independence of variance.
Significant differences were determined by one-way analysis of variance (ANOVA) based on the post
hoc Tukey test at the 5% level. Prior to ANOVA, the normality and homogeneity of variances were
tested by the Kolmogorov–Smirnov test and Levene's test, respectively. If normality was not met, log
or square-root transformation was implemented. One-way ANOVA was performed using SPSS 21.0
(SPSS Inc., Chicago, IL, USA).
Nonmetric multidimensional scaling (NMDS) analysis was used to describe and evaluate the
microbial community composition. Redundancy analysis (RDA) was performed to visualize the
associations between the microbial community composition and selected soil properties. The NMDS
and RDA were performed in the "Vegan" package of R (4.0.2). To describe the complex co-occurrence



patterns in various organisms, we constructed co-occurrence networks. We focused on the abundant
microbial phylotypes (with average relative abundance > 0.01% for bacteria and fungi) for network
construction. Nodes with Pearson correlations greater than 0.70 and $p < 0.05$ were retained. Network
visualization between microbial taxa was conducted by Gephi software. To obtain the keystone species
of each network, a *Zi-Pi* plot series was constructed to determine the role of each OTU. According to
Deng et al. (2012), the plot includes (a) peripheral nodes ($Z \leq 0.25$, $P \leq 0.62$). (b) module hubs ($Z >$
0.25, $P \leq 0.62$), (c) connectors ($Z \leq 0.25$, $P > 0.62$) and (d) network hubs ($Z > 0.25$, $P > 0.62$). From an
ecological perspective, OTUs in module hubs, connectors and network hubs may be regarded as the
microbial keystone taxa of the network systems.
A heatmap was constructed to reveal the associations between microbial traits and fertilizers, soil
properties, greenhouse emissions and ecosystem multifunctionality. The random forest algorithm was
performed in the R package "RandomForest" to estimate the importance predictors of soil properties
and microbial traits on ecosystem multifunctionality.

**3. Results**
**3.1 Cropland ecosystem services**
Data collection after a continuous 4-year in situ field experiment under different N input levels
revealed changes in cropland ecosystem services (Fig. 1). In terms of soil fertility, compared with the
N-limitation treatments (PK and 0.5N+PK), the SOC and total P contents were increased significantly
by the N+PK and 0.75N+PK treatments (Fig. 1a, c) ($P < 0.05$), while there were no significant changes
in the total N content (Fig. 1b). After straw decomposition (Fig. 1d), the amounts of straw C (Fig. 1e)
and N (Fig. 1f) released showed different responses to varying N fertilizer input levels. Generally,





N-rich treatments (N+PK and 0.75N+PK) significantly increased the straw decomposition rate and
achieved higher amounts of straw C and N release than the N-limitation treatments ($P < 0.05$).
However, there was no significant difference between N+PK and 0.75N+PK. Microbial biomass and
function were also sensitive to N fertilizer application (Fig. 1g, h, i and j). The MBC (Fig. 1g) and
MBN (Fig. 1h) contents were significantly higher in the N-rich treatments than in the other treatments.
However, the highest cellulase activity was observed in the 0.75N+PK treatment, which was
significantly higher than that in the other treatments (Fig. 1i) ($P < 0.05$), and the
N-acetyl-D-glucosaminidase activity decreased with the reduction in N application (Fig. 1j).
For the ecosystem dis-services (greenhouse gas emissions), with the increase in N fertilizer
application levels, $CO_2$ and $N_2O$ emissions gradually increased (Fig. 1k, m). No significant difference
was observed in $CH_4$ emissions under the different fertilization treatments (Fig. 1l). In addition, the N
fertilizer levels also had a strong influence on maize yields and aboveground biomass (Fig. 1n, o). Our
results indicated that the N+PK and 0.75N+PK treatments resulted in higher maize yields and
aboveground biomass than the other treatments ($P < 0.05$), suggesting that a 25% N fertilizer reduction
could be satisfactory for maize growth. As expected, the 0.75N+PK treatment achieved the highest
multifunctionality index (0.61), followed by N+PK (0.32), 0.5N+PK (-0.34) and PK (-0.59) (Fig. 1p).
However, although the 0.75N+PK treatment increased the straw N release amount and may meet
the requirements for plant growth, the total N input was still dominated by inorganic N input (Fig. S1).
Therefore, the N released from the straw cannot offset the deficiency of N fertilizer. Additionally,
contrasting N fertilizer input levels significantly changed the stoichiometry of C, N and P (Fig. S2).
Notably, the 0.75N+PK treatment significantly increased the C:N ratio compared with the 0.5N+PK
and PK treatments ($P < 0.05$). The lowest C:N ratio was shown for the 0.5N+PK treatment (Fig. S2a).



The N:P and C:P ratios showed no significant difference regardless of nutrient excess or limitation (Fig.
S2b and c).

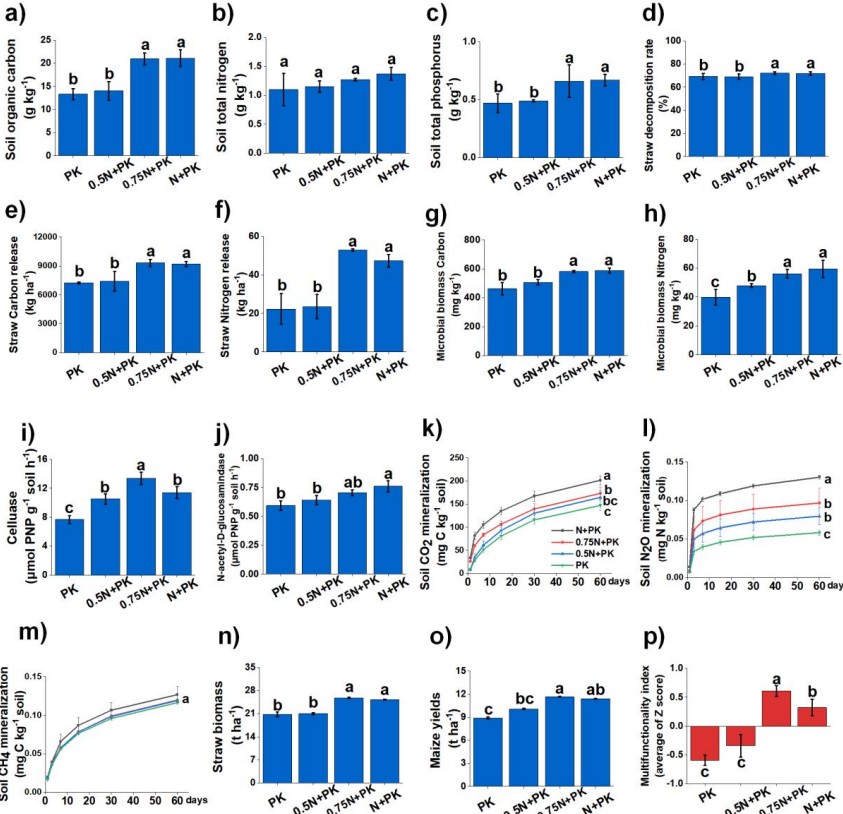


**Fig. 1** The 15 cropland variables and multifunctionality index under different N input levels after straw

return. Abbreviations: N+PK, straw return plus regular inorganic N-P-K fertilizers; 0.75N+PK, straw
return plus regular inorganic P-K with 25% N fertilizer reduction; 0.5N+PK, straw return plus regular
inorganic P-K with 50% N fertilizer reduction; PK, straw return plus regular inorganic P-K without N
fertilizer.
**3.2 Abundances of bacteria, fungi and genes encoding cellulose-degrading enzymes**
N fertilizer input levels had marked impacts on the abundances of fungi and bacteria (Table S3).
The highest fungal abundance was observed in the 0.75N+PK treatment, which was significantly



higher than that in the other treatments ($P < 0.05$). The N+PK treatment significantly increased
bacterial abundance compared with the PK treatment ($P < 0.05$), while there were no obvious
differences among the N+PK, 0.75N+PK and PK treatments. The ratios of fungi to bacteria also
showed contrasting responses to N fertilization (Table. S3). The 0.75N+PK treatment significantly
increased the ratio of fungi to bacteria compared with the other treatments ($P < 0.05$), and the lowest
ratio of fungi to bacteria was found in the PK treatment.
**Table 1 The abundances of genes encoding ellulose-degrading enzymes**
**across different N fertilizer level treatments after straw return**

| Treatment | *cbhI* gene abundance (×10⁶ copies g⁻¹ soil) | *GH48* gene abundance (×10⁷ copies g⁻¹ soil) | *cbhI*: *GH48* ratio |
|---|---|---|---|
| N+PK | 4.75±0.16 a | 1.68±0.01 a | 0.28±0.01 a |
| 0.75N+PK | 4.95±0.19 a | 1.60±0.04 a | 0.31±0.02 a |
| 0.5N+PK | 4.01±0.12 b | 1.54±0.08 a | 0.26±0.03 b |
| PK | 3.76±0.13 b | 1.40±0.06 b | 0.27±0.02 b |

The results show means ± standard deviations (n = 3). Different lowercase letters after values
indicate significant differences between each treatment, $P < 0.05$. N+PK, straw return plus
regular inorganic N-P-K fertilizers; 0.75N+PK, straw return plus regular inorganic P-K with
25% N fertilizer reduction ; 0.5N+PK, straw return plus regular inorganic P-K with 50% N
fertilizer reduction; PK, straw return plus regular inorganic P-K without N fertilizer.

N fertilizer input levels led to changes in the expression levels of genes encoding
cellulose-degrading enzymes (Table 1). The N-rich treatments achieved higher fungal *cbhI* and
bacterial *GH48* gene abundance than the N-limitation treatments. In contrast, the highest *cbhI* gene



abundance was shown in the 0.75N+PK treatment, while the highest *GH48* gene abundance was shown
in the N+PK treatment. Compared with the PK treatment, the ratio of the fungal *cbhI* gene to the
bacterial *GH48* gene increased significantly under the 0.75N+PK treatment ($P < 0.05$).

**3.3 Co-occurrence network analysis of the microbial community**
Regarding fungal alpha diversities, there were no significant differences in the Chao1 index across
treatments. The N+PK treatment significantly increased fungal richness compared with the PK
treatment ($P < 0.05$) (Table S4). In addition, the PK treatment resulted in lower bacterial richness than
the other treatments ($P < 0.05$). No significant difference was observed in the bacterial Chao1 index
across treatments (Table S4). NMDS plots showed that diverse N input levels significantly changed the
fungal (Fig. S3a) and bacterial communities (Fig. S3b) ($P < 0.05$).
We further conducted network analysis to identify co-occurrence patterns between specific
microbial taxa (Fig. 2). The cooccurrence network was aggregated into smaller coherent modules that
were examined to determine important module-trait relationships. The present network comprised 1963
nodes (composed of 1520 bacterial taxa and 443 fungal taxa) and 62206 edges with 52.49% positive
associations (Fig. 2a). The results showed four dominant ecological modules (1-4) that strongly
co-occurred within the multitrophic network, which contributed 86.10% of the whole network. Among
the four modules, bacteria accounted for the highest proportion in each module, contributing more than
70% of the total (Fig. 2b). The percentage of edges linking bacteria to bacteria (B-B) was higher than
that linking fungi to fungi (F-F) and bacteria to fungi (B-F). The highest proportion of B-B (80.32%)
was found in Module 3, while the highest proportion of B-F (32.66%) and F-F (6.00%) was found in
Module 4 (Fig. 2c).

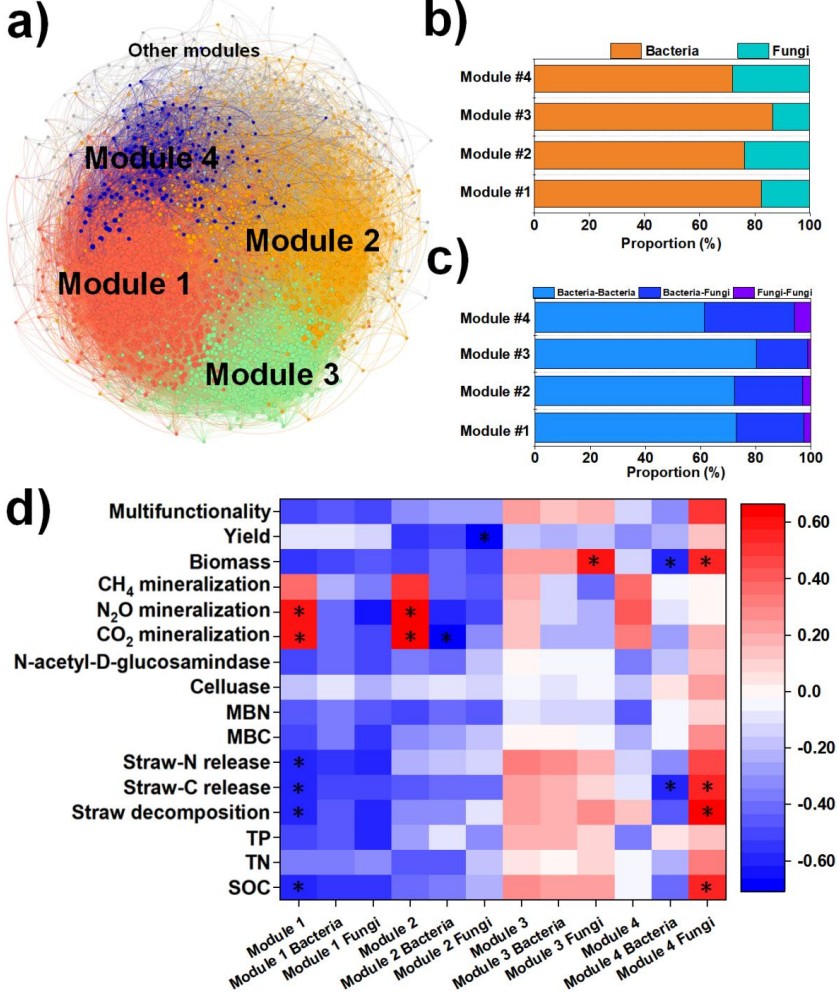


**Fig. 2** The relationships of microbial module communities with soil ecosystem services and

dis-services. Multitrophic network including multiple ecological modules. The colours of the

nodes represent different ecological modules (a). OTU number proportions of bacteria and fungi

(b). The proportions of the edges linking bacteria to bacteria (B-B), bacteria to fungi (B-F) and

fungi to fungi (F-F) in the major ecological modules (c). Links between the specific module

communities with soil ecosystem services and dis-services (d). * indicates significance at $P < 0.05$.

Abbreviations: SOC, soil organic carbon; C: N, the ratio of the SOC content to the total N content;



N: P, the ratio of the total N content to the total P content.

Individual nodes represented different roles in the microbial network based on the intramodule

connectivity $Zi$ and the intermodule connectivity $Pi$. ZP plots were constructed to identify the
topological roles of each node in the network (Fig. 3a). As shown in Fig. 3b, 113 microbial taxa (81
bacterial species and 32 fungal species) were regarded as connectors, and 43 microbial taxa (39
bacterial species and 4 fungal species) were regarded as module hubs. Specifically, module 2 (54)
contained the most keystone taxa, followed by module 1 (38) and module 3 (32).

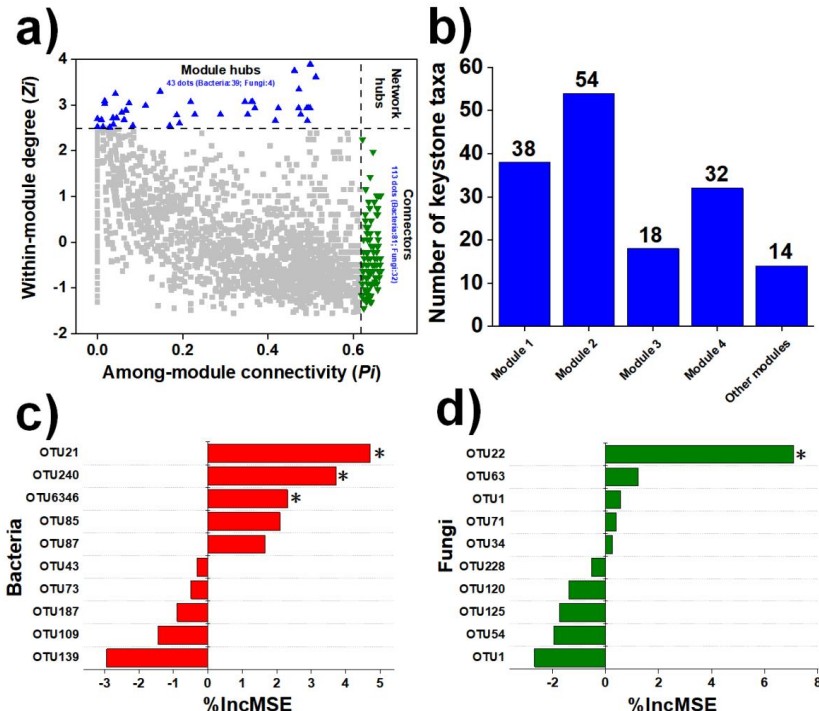


**Fig. 3** The topological roles of microbial taxa and their effect on the soil multifunctionality index.
The topological role of each OTU was determined according to the scatter plot of within-module
connectivity (Z) and among-module connectivity (P) (a). The distribution of keystone taxa in each
ecological module (b). Contribution of bacterial (c) and fungal OTUs (d) to the soil





multifunctionality index. *, ** and *** indicate significance at $P < 0.05$, 0.01 and 0.001,
respectively.

**3.4 Linkage between microbial traits and soil ecosystem multifunctionality**

The heatmap assumed close correlations between fertilizers (N input and straw), as well as soil
stoichiometry, and microbial traits (Fig. 4). Overall, the N input level, straw biomass and C:N ratio
upregulated the abundance of genes encoding cellulose-degrading enzymes. In addition, N input was
positively correlated with bacterial abundance, while a significant correlation was observed between
straw biomass and the N input level. The random forest model was also used to identify abiotic and
biotic attributes correlated with soil ecosystem multifunctionality (Fig. 5). The model explained
83.89% of the variance in ecosystem multifunctionality. The results indicated that the N input level,
straw biomass and soil C:N ratio were the most prominent abiotic factors affecting the ecosystem
multifunctionality index, while some biotic factors, such as the abundance of genes encoding
cellulose-degrading enzymes, significantly affected the ecosystem multifunctionality index.

Moreover, to clarify the potential main specific drivers of soil ecosystem services, correlations
between the microbial physiological traits and soil properties were determined to illuminate the role of
the microbial community in soil ecosystem multifunctionality (Fig. 3d). The results indicated that the
particular microbial module community was significantly correlated with soil ecosystem services. The
communities of modules 1 and 2 and the fungal community in module 4 showed potential in soil
ecosystem services (Fig. 3d). Specifically, significant correlations were observed between the SOC
content, straw decomposition, straw C/N release, $CO_2/N_2O$ mineralization and the module 1
community; the module 2 community was positively correlated with greenhouse gas emissions (except



for CH$_4$); and the fungal community in module 4 was positively correlated with the SOC content, straw
decomposition, straw C/N release and straw biomass. Furthermore, the bacterial and fungal
communities belonging to module 2 and the fungal community belonging to module 3 were
significantly correlated with CO$_2$ emission, maize yield and straw biomass.

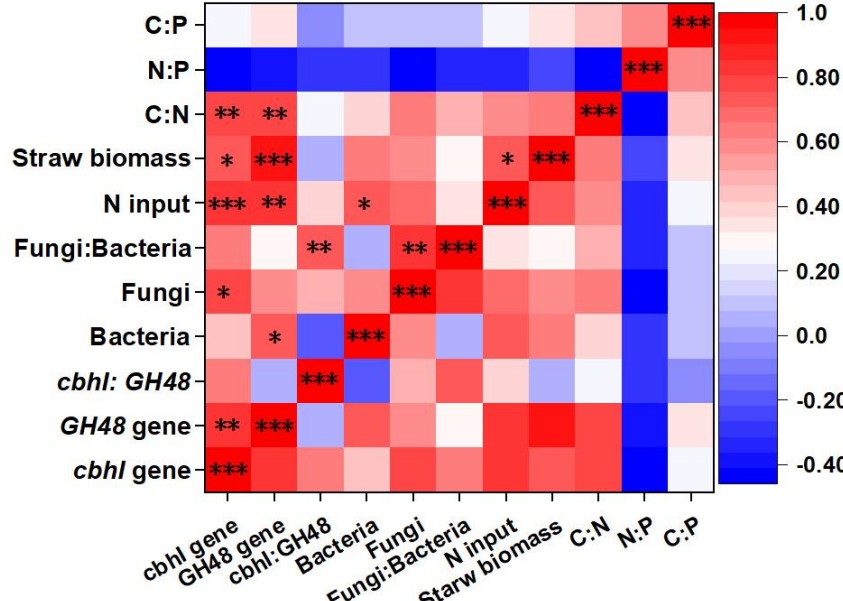


**Fig. 4** Heatmap revealing the correlation coefficients between microbial traits with fertilization
and soil stoichiometry. *, ** and *** indicate significance at $P < 0.05$, 0.01 and 0.001,
respectively. Abbreviations: C: N, the ratio of the SOC content to the total N content; N: P, the
ratio of the total N content to the total P content.

At the scale of microbial species, we selected the 20 keystone taxa (10 bacterial and 10 fungal
taxa) with the highest relative abundance for further analysis. The random forest models indicated that
the specific keystone taxa strongly influenced soil ecosystem multifunctionality (Fig. 4c and d).





Bacterial OTU21 (in module 1), OTU240 (in module 2) and OTU6346 (in module 3) were highlighted
as essential predictors of soil ecosystem multifunctionality, and fungal OTU22 (module 3) was also
found to be an important variable for predicting its changes. Subsequently, the relative abundances of
selected keystone taxa were different across different N fertilizer level treatments after straw return
(Table S5). The relative abundances of fungal OTU22 and bacterial OTU21 were higher in the N-rich
treatments than in the N-limitation treatments. Moreover, compared with the N+PK treatment, the
0.75N+PK treatment increased the relative abundances of fungal OTU22 by 38.20% and bacterial
OTU21 by 40.63%.

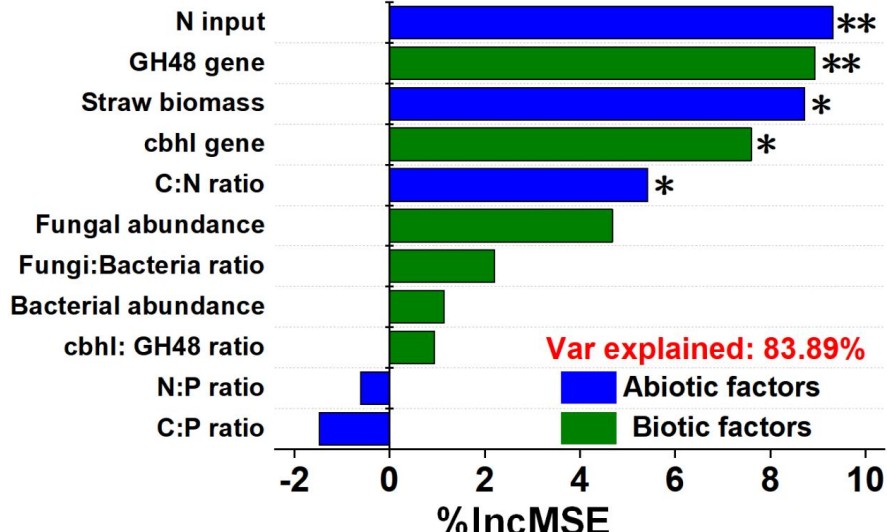


**Fig. 5** Contribution of abiotic and biotic variables to the soil multifunctionality index. *, ** and
*** indicate significance at *P* < 0.05, 0.01 and 0.001, respectively. Abbreviations: C: N, the ratio
of the SOC content to the total N content; N: P, the ratio of the total N content to the total P
content.



**4. Discussion**
**4.1 Effect of N fertilizer reduction on cropland ecosystem services after straw return**
Soil fertility, straw decomposition, C and N release amounts, and crop productivity showed an
overall positive effect with the increase in the N fertilizer input level and together contributed to
ecosystem services. Moreover, high N applications may also cause ecosystem dis-services due to the
surge in greenhouse gas emissions (Fig. 1). Our results indicated that the soil fertility index (SOC, total
N and P contents) increased under N-rich treatments as a result of high net primary production, in
accordance with previous reports (Liu et al., 2010; Williams et al., 2013). Higher microbial biomass C
and N, as well as relevant enzyme activities, were also observed under N-rich treatments, indicating the
strong positive impact of abundant N fertilizer application (Fig. 1g, h, i, j). It was reported that straw
return with N fertilizer application can stimulate microbial activity and promote biomass accumulation
(Treseder, 2008). The substantially increased straw decomposition and straw C and N release under
N-rich treatments may be primarily attributed to the activation of microbial activity (Fig. 1d, e, f),
which is consistent with previous research (Ramirez et al., 2012). Recent studies have also proven that
rational N input can stimulate microbial ex vivo production of extracellular enzymes to accelerate
straw decomposition and nutrient transformation (Chen et al., 2016). Moreover, it is well known that
fungi have high nutrient utilization efficiency; thus, more straw-derived C and N would be stored in
soil under N-rich treatments than under N-limited treatments (Hou et al., 2020). Higher N availability
is also the premise of straw decomposition and SOM formation due to the microbial "stoichiometry
decomposition" theory, while the "N-mining" theory in N-limitation treatments reveals that
oligotrophic species (such as K-strategists) degrade native SOM because of the lack of N fertilizer
inputs (Chen et al., 2014). Finally, the increases in SOC, total N, and P contents and straw C and N



release, as well as microbial biomass and function, are commonly attributed to high aboveground
biomass and maize yields (Fig. 1n, o), which are favourable from the viewpoint of ecosystem services.

However, the overuse of N inputs also causes ecosystem dis-services, such as unintended

environmental consequences (Tang et al., 2019). In the present study, greenhouse gas emissions were
quantified to evaluate the ecosystem dis-services under different N fertilizer input levels (Fig. 1k, l, m).
Straw return with N fertilizer addition might be the crucial driver of $CO_2$ and $N_2O$ emissions from
agroecosystems and has been widely studied in previous literature (Gregorich et al., 2005). $CO_2$ and
$N_2O$ emissions increased significantly compared with those under the PK treatment, likely by
stimulating the activity of copiotrophs when sufficient C and N substrates were provided. For example,
Dieleman et al. (2010) implied that N fertilizer addition significantly increased $CO_2$ and $N_2O$ by
increasing bacterial abundance through meta-analysis and field experiments, respectively. Qiu et al.
(2019) indicated that the emission of $CO_2$ enhanced root and mycorrhizal N uptake and increased $N_2O$
emissions, which was related to the changes in the soil denitrifier community composition in favour of
$N_2O$-producing taxa (nirK- or nirS-type). In addition, there was no difference in $CH_4$ emissions among
treatments, although contradictory results have been widely reported in previous literature (Tang et al.,
2019). Mapanda et al. (2011) and Liu et al. (2012) indicated that the emission of $CH_4$ depended highly
on the soil water content in maize crops, which is in line with our results.

In summary, compared with the N+PK treatment, the 0.75N+PK treatment supported multiple

ecosystem services, including promoting soil fertility, straw nutrient release and microbial activity and
alleviating greenhouse gas emissions (Fig. 1p). Therefore, a reduction of 25% in chemical N fertilizer
input with straw return may be the appropriate regime to promote ecosystem services in meadow soils
on the Northeast China Plain.



**4.2 Responses of the microbial composition and function to straw return with N fertilizer**
**reduction**
Fungal and bacterial abundances, as well as the ratio of fungi to bacteria, were sensitive to the
changes in the N fertilizer input levels (Table S3 and Fig. 2). Straw addition with N fertilizer input
supplied enough C and N for microbial metabolism, thus promoting microbial proliferation (Chen et al.,
2016). Generally, bacterial abundance decreased with reduced N fertilizer input. This is mainly because
bacteria are more sensitive to N availability than fungi, which is in line with a previous study (Ramirez
et al., 2020). Interestingly, it is worth noting that a 25% reduction of N fertilizer significantly increased
fungal abundance compared with regular N inputs. This result might be attributed to the negative effect
of excess N fertilizer (Wan et al., 2015). Moreover, Ning et al. (2020) demonstrated that the C:N ratio
was the pivotal factor in fungal community compositions after performing 7 long-term field
experiments under different fertilization conditions across China and reported a significant positive
correlation between them. Therefore, the 0.75N+PK treatment with a higher C:N ratio may facilitate
the proliferation of microorganisms and promote an increase in microbial abundance.
Subsequently, our results showed that N-rich treatments resulted in higher microbial
cellulose-degrading gene abundances than the PK treatment (Table 1), which demonstrated the
irreplaceable role of N inputs in straw degradation (Zhang et al., 2017). Additionally, compared with
bacterial *GH48* gene abundance, the increase in fungal *cbhI* gene abundance required adequate N
fertilizer inputs and was regulated by the soil C:N ratio, which suggests that rational N fertilizer inputs
could promote fungal function for further degradation of recalcitrant straw components (Hou et al.,
2020). Therefore, the ratio of *cbhI* gene abundance to *GH48* gene abundance was higher under
0.75N+PK than under the N-limitation treatments since the increased expression of a fungal



cellulose-degrading gene implies more straw C and N release.

Our results indicated that adequate N fertilizer upregulated fungal and *cbhI* gene abundances,

which may lead to multiple ecosystem services. It is therefore necessary to further explore the potential
associations between microbial traits and ecosystem services under diverse N fertilizer input levels.
**4.3 Linkages of cropland ecosystem services with microbial traits**

To clarify the effect of abiotic and biotic factors on soil ecosystem services, we then quantified the

contributions of abiotic and biotic attributes to the ecosystem multifunctionality index across N input
treatments (Fig. 4 and 5). Biotic factors, such as *cbhI* and *GH48* gene abundances, as well as abiotic
factors, including the C:N ratio, straw biomass and N input level, are also pivotal regulators of
ecosystem multifunctionality (Fig. 5). In general, promoting the rapid degradation of straw is an
important way to convert straw-C into SOM, thus improving soil fertility, aboveground biomass and
crop yield. In addition, fungi have a higher C utilization efficiency than bacteria; thus, a high fungal
*cbhI* gene abundance may achieve better soil multifunctionality (Hou et al., 2020). For abiotic factors,
the soil C:N ratio, straw biomass and N fertilizer input are always regarded as the main indicators of
soil fertility and health, likely due to providing various nutrient accessibilities and influencing the
microbial community composition (Ning et al., 2020).

Numerous studies have shown that core microbiota play a vital role in maintaining the stability of

soil microbial function and the complexity of microbial networks and then promoting soil nutrient
cycling ecosystem services (Ghannoum et al., 2015), and keystone species may show great explanatory
power in terms of specific network (or module) structure and function (Chen et al., 2019b). Heatmaps
and random forest models were used to illuminate the relationships of module communities with
ecosystem services (Fig. 2d and Fig 3c, d). In the present study, *Terrimonas* (bacterial species in



module 1) and *Lasiosphaeriaceae* (fungal species in module 3) were detected as the keystone taxa in
influencing soil multifunctionality of the cooccurrence network (Table S5). A previous study
demonstrated that straw addition significantly increased the relative abundance of *Lasiosphaeriaceae*,
which implied straw decomposition ability (Song et al., 2020). Afterwards, *Lasiosphaeriaceae* was
proven to promote straw-derived C and N accumulation by secreting multiple extracellular enzymes
(Guo et al., 2022). Meanwhile, Sun et al. (2023) revealed that *Lasiosphaeriaceae* abundance was
regulated by the soil C:N ratio, especially changes in mineral N. Therefore, *Lasiosphaeriaceae* can
effectively promote straw degradation and straw C and N release while driving the function and
community of module 1, which is consistent with our results (Fig. 2d). However, relatively few studies
have focused on the function of *Terrimonas*, so this study focused on *Chitinophagaceae*. As reported in
the previous literature, straw return was the main method to increase *Chitinophagaceae* abundance (Li
et al., 2021). Furthermore, *Chitinophagaceae* was indicated to have a strong ability to accumulate soil
C and N and degrade cellulose (Zhong et al., 2022), facilitating production improvement by regulating
the module 3 community and function, which is in line with our results (Fig. 2d).
Overall, straw return with sufficient N fertilizer application can increase the C:N ratio and
stimulate microbial traits, which ultimately achieve soil ecosystem multifunctionality (Fig. 6). Straw
return without enough N supply cannot support ecosystem services due to the decomposition of native
SOM and the out-of-balance microbial community composition, according to the "N-mining" theory
(Chen et al., 2014); straw return with sufficient N application (N+PK and 0.75N+PK) can promote soil
fertility, straw release, microbial activity and crop productivity, which can be explained by the
"stoichiometry decomposition" theory (Chen et al., 2014). Meanwhile, N+PK also caused more serious
ecosystem dis-services, such as greenhouse gas emissions, than the 0.75N+PK treatment. Moreover,



compared with the N+PK treatment, the 0.75N+PK treatment increased the soil C:N ratio and
stimulated microbial module 1 and 3 communities function, *cbhI* gene abundance, and keystone taxa
abundances, which were significantly positively correlated with soil ecosystem multifunctionality. Our
study provides evidence that a 25% reduction of chemical N fertilizer after straw return was the
optimal agronomic measure for ecosystem services in meadow soil on the Northeast China Plain.

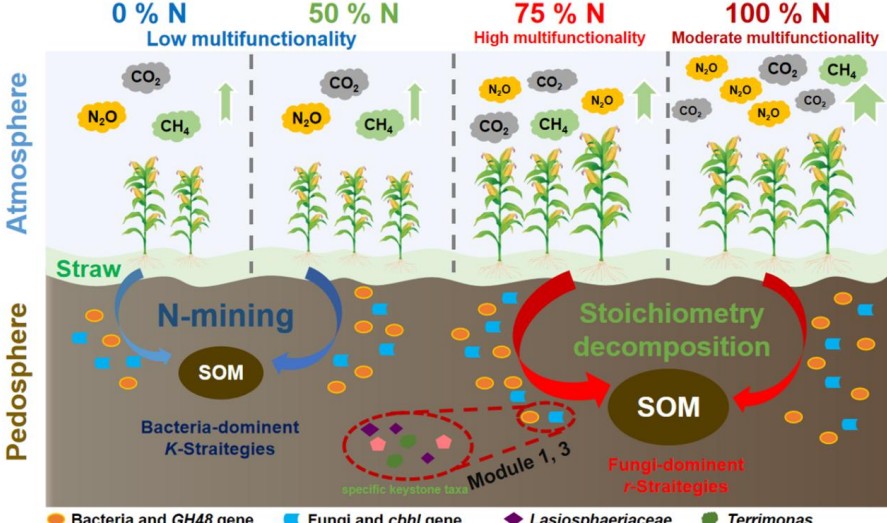


**Fig. 6** A graphical sketch of the changes in ecosystem services and potential microbial mechanisms in
response to different chemical N fertilizer application rates after straw return. N, nitrogen; SOM, soil
organic matter

**5. Conclusion**
Straw return combined with different chemical N fertilizer application rates significantly changed
ecosystem services and dis-services. Collectively, our work indicates that compared with the N+PK
treatment, straw return with a 25% reduction in chemical N fertilizer has the potential to improve
ecosystem services by maintaining soil fertility, productivity, microbial biomass and function,



promoting straw decomposition and C and N release and alleviating greenhouse gas emissions. The
0.75N+PK treatment achieved higher soil ecosystem multifunctionality than all other treatments. In
addition, the N input level, straw biomass and soil C:N ratio can upregulate the abundances of the *cbhI*
and *GH48* genes, which may together achieve soil ecosystem multifunctionality.

Furthermore, the changes in multiple soil ecosystem services were strongly associated with

microbial module communities and keystone taxa. The relationships between ecosystem services and
microbial traits were examined here to confirm that the *Lasiosphaeriaceae* driving the function and
structure of the module 1 community leads to the promotion of straw degradation and straw C and N
release, while *Terrimonas* driving the function and structure of the module 3 community probably
contributes to production improvement under 0.75N+PK treatment. Therefore, a 25% reduction in
chemical N fertilizer with straw return might be a win–win strategy that not only produces considerable
ecological benefits for the pedosphere and atmosphere but also reduces fertilizer expenditures in
meadow soil on the Northeast China Plain.



**Declaration of competing interests**

The authors declare that they have no known competing financial interests or personal relationships that could have appeared to influence the work reported in this paper.

**Acknowledgements**

We thank all our lab colleagues for their assistance with soil sampling and analyses. This work was jointly supported by the Anhui Postdoctoral Science Foundation (2022B638); the Special Project of Zhongke Bengbu Technology Transfer Center (ZKBB202103); China Postdoctoral Science Foundation (2023M733542); Special Research Assistant Project of Chinese Academy of Sciences (2023000140); and Chinese Academy of Sciences (CASHIPS) Director' s Fund (YZJJ2023QN37).

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





**Figure captions**

**Fig. 1** The 15 cropland variables and multifunctionality index under different N input levels after straw

return. Abbreviations: N+PK, straw return plus regular inorganic N-P-K fertilizers; 0.75N+PK, straw

return plus regular inorganic P-K with 25% N fertilizer reduction; 0.5N+PK, straw return plus regular

inorganic P-K with 50% N fertilizer reduction; PK, straw return plus regular inorganic P-K without N

fertilizer.

**Fig. 2** The relationships of microbial module communities with soil ecosystem services and

dis-services. Multitrophic network including multiple ecological modules. The colours of the

nodes represent different ecological modules (a). OTU number proportions of bacteria and fungi

(b). The proportions of the edges linking bacteria to bacteria (B-B), bacteria to fungi (B-F) and

fungi to fungi (F-F) in the major ecological modules (c). Links between the specific module

communities with soil ecosystem services and dis-services (d). * indicates significance at $P < 0.05$.

Abbreviations: SOC, soil organic carbon; C: N, the ratio of the SOC content to the total N content;

N: P, the ratio of the total N content to the total P content.

**Fig. 3** The topological roles of microbial taxa and their effect on the soil multifunctionality index.

The topological role of each OTU was determined according to the scatter plot of within-module

connectivity ($Z$) and among-module connectivity ($P$) (a). The distribution of keystone taxa in each

ecological module (b). Contribution of bacterial (c) and fungal OTUs (d) to the soil

multifunctionality index. *, ** and *** indicate significance at $P < 0.05$, 0.01 and 0.001,

respectively.

**Fig. 4** Heatmap revealing the correlation coefficients between microbial traits with fertilization

and soil stoichiometry. *, ** and *** indicate significance at $P < 0.05$, 0.01 and 0.001,





respectively. Abbreviations: C: N, the ratio of the SOC content to the total N content; N: P, the

ratio of the total N content to the total P content.

**Fig. 5** Contribution of abiotic and biotic variables to the soil multifunctionality index. *, ** and

*** indicate significance at $P$ < 0.05, 0.01 and 0.001, respectively. Abbreviations: C: N, the ratio

of the SOC content to the total N content; N: P, the ratio of the total N content to the total P

content.

**Fig. 6** A graphical sketch of the changes in ecosystem services and potential microbial mechanisms in

response to different chemical N fertilizer application rates after straw return. N, nitrogen; SOM, soil

organic matter





**Table 1 The abundances of genes encoding ellulose-degrading enzymes across different N fertilizer level treatments after straw return**

| Treatment | cbhI gene abundance ($\times 10^6$ copies g$^{-1}$ soil) | GH48 gene abundance ($\times 10^7$ copies g$^{-1}$ soil) | cbhI: GH48 ratio |
|---|---|---|---|
| N+PK | 4.75±0.16 a | 1.68±0.01 a | 0.28±0.01 a |
| 0.75N+PK | 4.95±0.19 a | 1.60±0.04 a | 0.31±0.02 a |
| 0.5N+PK | 4.01±0.12 b | 1.54±0.08 a | 0.26±0.03 b |
| PK | 3.76±0.13 b | 1.40±0.06 b | 0.27±0.02 b |

The results show means ± standard deviations (n = 3). Different lowercase letters after values indicate significant differences between each treatment, P < 0.05. N+PK, straw return plus regular inorganic N-P-K fertilizers; 0.75N+PK, straw return plus regular inorganic P-K with 25% N fertilizer reduction ; 0.5N+PK, straw return plus regular inorganic P-K with 50% N fertilizer reduction; PK, straw return plus regular inorganic P-K without N fertilizer.