# Peer review of "Title: Straw return with diverse nitrogen fertilizer application rates modulate ecosystem services"

_EGUsphere, 2023_

## Referee Comment (RC2)

[referee-annotated manuscript omitted]

---

## Community Comment (CC2)

[revised manuscript text omitted]

---

## Author Comment (AC1)

List of responses

Dear Editor and Reviewers:

Thank you for the reviewers' comments concerning our manuscript entitled "Moderate N fertilizer reduction with straw return modulates ecosystem services and microbial traits in a meadow soil" (Manuscript ID No. egusphere-2023-2498). These comments were all valuable for improving our manuscript and provided important guidance for our research. We have studied the comments carefully and have made corrections that we hope will meet with your approval. The main corrections in the paper and the responses to the reviewer's comments are as follows:

General Review: This study investigates the impacts of different N fertilizer rates on maize through a four-year field experiment, exploring its effects on ecosystem functions such as soil fertility, straw degradation, greenhouse gas emissions, and maize yield. The authors introduce the terms "soil multifunctionality" and "multiple ecosystem services" in a small-scale experiment, when in fact these terms are used in large-scale studies, generally in several locations and with many biological replicates. Also, the manuscript introduces the term "ecosystem dis-services" to represent greenhouse gas emissions, a usage that is relatively uncommon. Furthermore, the research focusses too much on "straw return" without including a treatment group without straw return, complicating the ability to draw robust conclusions about its effects on the experiment. The study lacks essential innovation as the impact of varying N fertilizer rates on soil fertility and greenhouse gas emissions is well studied. Despite this, the manuscript provides valuable insights, such as the revelation that reducing N fertilization by 25% yields comparable results to conventional N application while simultaneously decreasing greenhouse gas emissions. Furthermore, the study highlights increased straw decomposition and N mineralization with a 25% N reduction, which was quantified through the assessment of functional genes (cbhI and GH48) associated with fungi and bacteria, respectively. Regrettably, in its current form, I cannot endorse the publication of this manuscript in SOIL. I recommend that the authors reshape the manuscript's perspective, emphasizing the robust findings,

avoiding the use of "soil multifunctionality," and consider submitting a completely revised version.

Specifics comments:

The introduction section is too extensive and it should be reduced to maximum 2.5 pages with a double space.

1.  The introduction section is too extensive and it should be reduced to maximum 2.5 pages with a double space.

**Reply:** Thank you for the comment. We have reduced the Introduction section.

2.  Line 33: How could this be an N deficiency effect since the 0.75N+PK treatment had a slightly better corn yield than the N+PK treatment and the same amount of straw biomass?

**Reply:** Thank you for the comment. We have revised this sentence.

The 0.75N+PK treatment had overall positive effects on soil fertility, productivity, straw decomposition, and microbial abundance and function and alleviated greenhouse gas emissions. (Lines 32-33)

3.  Lines 73-74: Please, explain what is an "opaque environment".

**Reply:** Thank you for the comment. Originally, "opaque environment" meant "invisible to the naked eye". According to your comments above, the sentence has been deleted.

4.  Lines 153-154: It's important to address the exact time that the litter bags were left in the field.

**Reply:** Thank you for the comment. We have added the necessary information.

On May 2$^{nd}$, 2021, litter bags containing straw were buried at 10 cm depth in a random design to prevent bags associated with a given decomposition stage from being placed together. The litter bags were collected after harvest on October 1$^{st}$, 2021. (Lines 139-141)

5. Lines 209-211: The authors should specify the reference used to calculate the soil multifunctionality.

**Reply:** Thank you for the comment. We have added the reference.

6. Lines 212-213: The method description indicates that N2O and CO2 emissions were given the same weight as other soil attributes. This means that higher emissions contribute positively to soil multifunctionality. Therefore, the reader may be confused by thinking that higher greenhouse gas emissions generate beneficial effects on the environment, whereas this effect is exactly the opposite. I suggest the authors review how they work with "soil multifunctionality".

**Reply:** Thank you for the comment. In this study, greenhouse gas emissions had a negative effect on soil multifunctionality. We have added relevant descriptions to the Materials and methods.

Notably, the opposite numbers of greenhouse gas emissions were used to evaluate their negative effects. (Lines 198-199)

7. Line 218: What is a success of DNA extraction? The agarose gel in a electrophoresis can evaluate the DNA integrity, i.e., if the DNA is fragmented or not.

**Reply:** Thank you for the comment. We have revised this sentences.

The quality of the extracted DNA was characterized by electrophoresis on 1% (wt/vol) agarose gel. (Lines 206-207)

8. Lines 227-228, 238-239: The PCR stages should be addressed correctly as stage of DNA denaturation, repeated cycles of DNA annealing and the final stage of extension.

**Reply:** Thank you for the comment. We have revised these sentences.

The thermal qPCR profiles for bacteria and fungi were as follows: 95 °C for 2 min for DNA denaturation, 35 cycles of (95 °C for 30 s, 60 °C for 30 s, 72 °C for 30 s, and 80 °C for 15 s) for DNA annealing, and 81 °C for 10 s for DNA extension; and 95 °C for 10 min for DNA denaturation, 40 cycles of (95 °C for 15 s, 52 °C for 30 s, 72 °C for 30 s, and 79 °C for 30 s) for DNA annealing, and 81 °C for 10 s for DNA extension, respectively. (Lines 214-218)

The qPCR thermal profiles for the target genes of *GH48* and *cbhI* were as follows: 95 °C for 5 min for DNA denaturation, 40 × (94 °C for 30 s, 60 °C for 45 s, and 72 °C for 90 s) for DNA annealing, and 84 °C for 10 s for DNA extension; and 94 °C for 4 min for DNA denaturation, 40 × (94 °C for 45 s, 50 °C for 30 s, and 72 °C for 60 s) for DNA annealing, and 81 °C for 10 s for DNA extension, respectively. (Lines 226-230)

9. Lines 282-283: There are no RDA results in the main manuscript or in the supplementary material.

**Reply:** Thank you for the comment. We have deleted the mention of RDA.

10. Lines 294-295: The heatmap was based on a correlation method? Which one?

**Reply:** Thank you for the comment. We have revised this sentence.

A first heatmap was constructed to reveal the associations between soil ecosystem services and microbial module communities. Another heatmap was constructed to reveal the associations between microbial traits and fertilizers, soil properties, greenhouse emissions and ecosystem multifunctionality. (Lines 286-288)

11. Lines 309-310: Please explain which microbial function. Also, the microbial biomass was sensitive for what? Microbial biomass was increased or reduced? What are the magnitude changes of your treatments?

**Reply:** Thank you for the comment. We have revised this sentence.

The MBC and MBN contents, as well as the associated enzyme activities, changed

after the application of different N fertilizer rates. (Lines 302-303)

12. Lines 315 and every time that the term "ecosystem dis-services" is used: I suggest the authors to use the correct terminology of "greenhouse gas emissions" instead of "ecosystem dis-services". This can prevent readers from making wrong associations with the authors' results and increase the manuscript visibility for readers looking for more information on N2O emissions under different rates of nitrogen fertilizer use.

**Reply:** Thank you for the comment. We have revised this sentence.

Regarding greenhouse gas emissions, with decreasing N fertilizer application levels, $CO_2$ and $N_2O$ emissions gradually decreased. (Lines 308-309)

13. Lines 319-21: This should be moved to the discussion section.

**Reply:** Thank you for the comment. We have deleted this sentence.

14. Line 332: Figures 1K, 1L, and 1M use the term "mineralization" whereas it should use the term "emission". I suggest the authors change it to avoid the reader's misunderstanding.

**Reply:** Thank you for the comment. We have revised the terms in Figure 1.

[Figure]

15. Line 367 and Figure S3 (supplementary material): The ANOSIM test was used in the NMDS plot and it should be included in the Material and Methods section.

**Reply:** Thank you for the comment. We have added the necessary content to the Materials and Methods.

Analysis of similarities (ANOSIM) was used to examine the significant differences in microbial community structure under different fertilization treatments. (Lines 273-275)

16. Lines 372-373: The multitrophic network seems to be an important result of the authors' co-occurrence analysis. Therefore, it should be included in the Material and Methods section.

**Reply:** Thank you for the comment. We have added the necessary content to the

Materials and Methods.

Network visualization of microbial taxa and ecological clusters of microbial phylotypes was conducted with Gephi software. (Lines 278-280)

17. Line 381: Please include the network topological properties to have a better overview of the number of nodes, edges and the proportion of positive and negative edges.

**Reply:** Thank you for the comment. We have added these properties.

[Figure]

18. Lines 416 and 419: Figure 3D shows the contribution of fungal OTUs to the soil multifunctionality index. Perhaps the authors made a mistake and referenced the

wrong figure. I suggest a review of this part of the results.

**Reply:** Thank you for the comment. We have corrected these faults.

Moreover, to clarify the potential main drivers of soil ecosystem services, the correlations between microbial physiological traits and soil properties were calculated to determine the role of the microbial community in soil ecosystem multifunctionality (Fig. 2d). The results indicated that the microbial module community was significantly correlated with soil ecosystem services. The communities of modules 1 and 2 and the fungal community in module 4 showed potential contributions to soil ecosystem services (Fig. 2d). (Lines 405-410)

19. Lines 433-434: Again, the figure reference is incorrect. I think this result may refer to Figure 3C and 3D. I suggest the authors review whether all results are addressed to their respective figures.

**Reply:** Thank you for the comment. We have corrected these faults.

We selected the 20 keystone taxa at the species level (10 bacterial and 10 fungal taxa) with the highest relative abundance for further analysis. The random forest model results indicated that specific keystone taxa strongly influenced soil ecosystem multifunctionality (Fig. 3c and d). (Lines 422-424)

20. Lines 451-492: Weak discussion on the effects of reducing the rates of N application. The discussion is focused too much on N application and not on different N rates.

**Reply:** Thank you for the comment. We have added some discussion.

Our results indicated that 0.75N+PK maintained parameters related to the soil fertility index and net primary production compared to N+PK. The results demonstrate that the effects of 0.75N+PK on soil ecosystem services are similar to those of N+PK. Therefore, it can be concluded that 0.75N+PK is a more efficient and effective option for improving soil ecosystem services. Moreover, 0.75N+PK may enhance N fertilizer use efficiency and stimulate microbial functioning by altering the stoichiometry of C, N and P in the soil, ultimately promoting soil fertility and crop yield (Liu et al., 2010).

Reducing the amount of N fertilizer by more than 50% led to insufficient N input to meet the needs of both crops and microbes, resulting in a decline in soil health (Williams et al., 2013). (Lines 446-454)

21. Lines 451-453: This sentence is confusing. The variables soil fertility", "C and N release" and "crop productivity" were defined as ecosystem services by the authors. How are these variables themselves are contributing to the increase in ecosystem services?

**Reply:** Thank you for the comment. We have revised this sentence.

Soil fertility, straw decomposition, C and N release amounts, and crop productivity were mostly higher under 0.75N+PK and N+PK than under the other treatments, implying that better soil multifunctionality was achieved. (Lines 438-440)

22. Lines 473-474: It is important to describe what are the "unintended consequences".

**Reply:** Thank you for the comment. We have revised this sentence.

However, excess N input also causes increased greenhouse gas emissions. (Line 467)

23. Line 493: There is no data from microbial composition (i.e., taxonomy) in this manuscript. It's better to change the title to "microbial abundance".

**Reply:** Thank you for the comment. We have revised this heading.

4.2 Responses of microbial abundance and function to straw return with N fertilizer reduction (Line 489)

24. Lines 502-505: What was the C:N ratio on the referenced paper? It's similar to the authors' results? It's crucial to compare the results from the authors with similar results found in the literature.

**Reply:** Thank you for the comment.

Gao et al. (2015) indicated that the optimal ratio of C to N inputs was 20:1, which

may meet the demands of maize growth and microbial proliferation. (Lines 500-502)

Therefore, the 0.75N+PK treatment with a higher C:N ratio (16.47) may facilitate the

proliferation of microorganisms and promote an increase in microbial abundance.

(Lines 509-510)

25. Line 506: I was expecting some discussion about the fungi:bacteria ratio since the
    0.75N+PK treatment showed a better result than the N+PK treatment.

**Reply:** Thank you for the comment.

Gao et al. (2015) indicated that the optimal ratio of C to N inputs was 20:1, which
may meet the demands of maize growth and microbial proliferation. It is well known
that fungi have a stronger C utilization efficiency than bacteria (Duan et al., 2021).
Therefore, increasing fungal abundance and lowering the ratio of bacteria to fungi are
crucial for straw degradation and SOC accumulation. Previous studies have shown
that the C:N ratio of fungi is greater than 20; however, the C:N ratio of bacteria is less
than 10. Excessive N fertilizer input may reduce the soil C:N ratio, while low N
fertilizer input cannot meet the growth requirements of crops and microorganisms
(Ning et al., 2020). Therefore, appropriate enhancement of the soil C:N ratio can
increase the ratio of fungi to bacteria, stimulate fungal function, and promote straw
degradation and SOC accumulation. (Lines 500-509)

26. Lines 516-517: What would be an "adequate N fertilizer"? It is better to inform
    that the 0.75N+PK treatment showed better results than the N+PK treatment.
    Moreover, the term "multiple ecosystem services" is not the most appropriate
    here since both the cbhI and GH48 genes are related to the straw degradation
    function.

**Reply:** Thank you for the comment. We have revised these sentences.

Our results indicated that 75%-100% N fertilizer could upregulate fungal and

*cbhI* gene abundances, which may lead to straw decomposition and SOC

accumulation. It is therefore necessary to further explore the potential associations between microbial traits and ecosystem services under varying N fertilizer input levels. (Lines 520-523)

27. Lines 534-535: It's better to discuss your main results and not the methods used in data analysis.

**Reply:** Thank you for the comment. We have deleted this sentence.

Thank you for your valuable and comments. We hope our responses will meet with your approval.

---

## Author Comment (AC2)

List of responses

Dear Editor and Reviewers:

Thank you for the reviewers' comments concerning our manuscript entitled "Moderate N fertilizer reduction with straw return modulates ecosystem services and microbial traits in a meadow soil" (Manuscript ID No. egusphere-2023-2498). These comments were all valuable for improving our manuscript and provided important guidance for our research. We have studied the comments carefully and have made corrections that we hope will meet with your approval. The main corrections in the paper and the responses to the reviewer's comments are as follows:

I think this is an interesting study and I carefully completed the review. The authors investigated the effects of N input levels on microbial community and agroecosystem services after straw return. And they tried to explain the relationship between N input and ecosystem services from two aspects: microbial genes abundance and module communities. This work can be accepted with the following modifications.

1. As straw was added in all the treatments in this research, the title"Moderate N fertilizer reduction with straw return modulates ecosystem services and microbial traits in a meadow soil" might be more proper.

**Reply:** Thank you for the comment. The title has been changed to "Moderate N fertilizer reduction with straw return modulates ecosystem services and microbial traits in a meadow soil". (Lines 1-2)

2. Abstract: It is recommended to add primary data support rather than purely verbal descriptions. And provide the results about other treatments to compare with 0.75N+PK.

**Reply:** Thank you for the comment. We have added specific descriptions of these data.

Specifically, no significant differences were observed in SOC, total N and P contents, straw C and N release amounts, microbial biomass C and N contents, or cellulase and

N-acetyl-D-glucosaminidase activities relative to those of 0.5N+PK and PK. Greenhouse gas mineralization was reduced with decreasing N input. Moreover, 0.75N+PK had the highest straw biomass and yield, which were significantly higher than those in 0.5N+PK and PK. (Lines 34-38)

3. L29-30 If possible, "N+PK" and "PK" can be changed as "1N+PK" and "0N+PK", which is more comparable and clear to express the significance of different treatments.

**Reply:** Thank you for the comment. In my opinion, the current version is clear enough and can be clearly distinguished from 0.9N+PK and 0.8N+PK.

4. In the introduction, please replenish relative introduction on reduction of N fertilization.

**Reply:** Thank you for the comment. We have added the necessary descriptions.

Recent research has indicated that an appropriate reduction in N fertilizer input can not only maintain crop yield by increasing N fertilizer use efficiency but also promote soil health by regulating the soil C:N ratio (Chen et al., 2014). However, an excessive reduction in N fertilizer input can lead to an "N-mining" effect, resulting in the loss of soil organic matter, which reduces crop yields (Chen et al., 2014). (Lines 58-62)

5. Line 33: "...... N deficiency." should be " ...... reduction of N fertiliser application ."

**Reply:** Thank you for the comment. We have revised this sentence.

The 0.75N+PK treatment had overall positive effects on soil fertility, productivity, straw decomposition, and microbial abundance and function and alleviated greenhouse gas emissions. (Lines 32-34)

6. Materials and methods, Supply references for all the determination methods.

**Reply:** Thank you for the comment. We have added the appropriate references to the Materials and methods.

7. In the results, the description of " with the increase of N fertiliser application" should be "with the reduction of N fertiliser application", and redescribe relative results"

**Reply:** Thank you for this comment. We have rewritten this sentence.

Regarding greenhouse gas emissions, with decreasing N fertilizer application levels, $CO_2$ and $N_2O$ emissions gradually decreased. (Lines 308-309)

8. What does mean of "straw biomass" in Fig4?

**Reply:** Thank you for this comment. "Straw biomass" refers to the dry weight of aboveground straw, which is explained in the Materials and methods. (Lines 184-185)

9. Lines 73-82: This paragraph is intended to express the role of microorganisms in ecosystem services, but needs to complement the examples of previous research.

**Reply:** Thank you for the comment. We have added some examples.

For example, Ning et al. (2020) found that long-term manure application increased the abundance of specific fungi involved in yield improvement. Duan et al. (2021) reported that adequate N input improved the cellulose degradation ability of bacteria. (Lines 79-81)

10. Line 112:It is suggested to add the purpose of this study.

**Reply:** Thank you for the comment. We have added the purpose.

The purpose of this study was to optimize the N fertilizer application rate to achieve soil ecosystem multifunctionality and explore the potential microbial mechanism in a Mollisol. (Lines 101-103)

11. Line 297: Add a related version of the R language.

**Reply:** Thank you for the comment. We have added the version of the R language used.

12. In discussion, I recommend supplementing the evidence on the relationship between straw C and N release and greenhouse gases. As we all know, the efficient conversion of straw C and N into SOM rather than CO2 and N2O is a key issue. Adding relevant content can greatly improve the quality of your manuscript.

**Reply:** Thank you for the comment. In this study, straw C and N release and greenhouse gas emissions did not occur in the same environment. Straw C and N release levels were determined in field experiments, while greenhouse gas emissions were calculated in cultivation experiments.

In addition, the two experimental periods were not consistent; straw C and N release were measured after approximately 5 months, and greenhouse gas emissions were measured after only 60 days.

Therefore, the relationship between these parameters was not discussed in this study. Your suggestion is very insightful, and relevant experiments will be carried out in subsequent research to reveal the relationships between them.

13. L403-404 It is not clear to show the description of Fig.4.

**Reply:** Thank you for the comment. We have rewritten the description.

The heatmap showed close correlations of fertilizers (N input and straw return) with soil stoichiometry and microbial traits. (Lines 395-396)

14. Lines 454-456:It is suggested to revise this sentence.

**Reply:** Thank you for the comment. We have revised this sentence.

Higher MBC and MBN values, as well as relevant enzyme activities, were also observed under the N-rich treatments, indicating the strong positive impact of abundant N fertilizer application. (Lines 440-442)

15. Line 500: Deleting "Interestingly"

**Reply:** Thank you for the comment. We have deleted this term.

Thank you for your valuable comments. We hope our responses will meet with your approval.

---

## Author Comment (AC3)

List of responses

Dear Editor and Reviewers:

Thank you for the reviewers' comments concerning our manuscript entitled "Moderate N fertilizer reduction with straw return modulates ecosystem services and microbial traits in a meadow soil" (Manuscript ID No. egusphere-2023-2498). These comments were all valuable for improving our manuscript and provided important guidance for our research. We have studied the comments carefully and have made corrections that we hope will meet with your approval. The main corrections in the paper and the responses to the reviewer's comments are as follows:

Duan et al. conducted a four-year field experiment to investigate the relationships between soil ecosystem services and microbial traits under varying rates of nitrogen (N) fertilizer application with straw return. The results indicated that a 25% reduction in chemical N fertilizer is optimal for enhancing ecosystem services. This study is interesting, and the findings contribute to nitrogen management following straw return. I have several suggestions to enhance the manuscript's quality:

1. Line 77, "To our knowledge, fertilization-induced changes in microbial communities and functions are fundamental to the regulation of a variety of ecosystem multifunctionalities", the importance of microbial community is widely acknowledged, so remove "To our knowledge". Please correct similar statement.

**Reply:** Thank you for the comment. We have deleted this term.

2. Line 126, (2) 25% reduction XXX; (3) XXX

**Reply:** Thank you for the comment. We have revised these items.

Four treatments were established with different N fertilizer input levels after straw return to the field for 4 years as follows: (1) regular chemical fertilization, N+PK (300 kg urea (N 46%) ha$^{-1}$ yr$^{-1}$, 250 kg diammonium phosphate (P$_2$O$_5$ 48%) ha$^{-1}$ yr$^{-1}$, 150 kg potassium chloride (K$_2$O 50%) ha$^{-1}$ yr$^{-1}$); (2) 25% reduction of N fertilizer, 0.75N+PK (225 kg urea ha$^{-1}$ yr$^{-1}$, 250 kg diammonium phosphate ha$^{-1}$ yr$^{-1}$, 150 kg

potassium chloride ha$^{-1}$ yr$^{-1}$); (3) 50% reduction of N fertilizer, 0.50N+PK (150 kg urea ha$^{-1}$ yr$^{-1}$, 250 kg diammonium phosphate ha$^{-1}$ yr$^{-1}$, 150 kg potassium chloride ha$^{-1}$ yr$^{-1}$); and (4) no N fertilizer, PK (250 kg diammonium phosphate ha$^{-1}$ yr$^{-1}$, 150 kg potassium chloride ha$^{-1}$ yr$^{-1}$). (Lines 111-118)

3. Line 138, 4 treatments with 3 replications each?

**Reply:** Thank you for the comment. We have revised the text.

In total, 12 soil samples were collected from the 4 treatments. Each treatment included 3 replicates. (Lines 128-129)

4. Lines 143-145, please clarify if it is rhizosphere soil or bulk soil.

**Reply:** Thank you for the comment. We have revised the text. The samples were all bulk soil samples.

One part of the bulk soil sample was air-dried to measure soil properties, and the other part was used for microbial molecular analysis. (Lines 132-133)

5. Line 200, the multifunctionality index is simply calculated by averaging the Z-scores of the 15 variables. There is a question, is more greenhouse gas emission better (also see 474-475)?

**Reply:** Thank you for the comment. We have revised the text. Increased greenhouse gas emissions negatively affect soil ecosystem multifunctionality. According to the calculations in the Materials and methods, the negative values of greenhouse gas emissions were used, as greater values indicate lower soil ecosystem multifunctionality.

We also added a precise description.

Notably, the opposite numbers of greenhouse gas emissions were used to evaluate their negative effects. (Lines 198-199)

In the present study, greenhouse gas emissions were quantified to evaluate the ecosystem dis-services under different N fertilizer input levels: the greater the emissions were, the lower the soil ecosystem multifunctionality was. (Lines 467-470)

6.  I would recommend the authors check the MS carefully, including English.

**Reply:** Thank you for the comment. We have checked it carefully.

Thank you for your valuable comments. We hope our responses will meet with your approval.

---

## Author Comment (AC4)

List of responses

Dear Editor and Reviewers:

Thank you for the reviewers' comments concerning our manuscript entitled "Moderate N fertilizer reduction with straw return modulates ecosystem services and microbial traits in a meadow soil" (Manuscript ID No. egusphere-2023-2498). These comments were all valuable for improving our manuscript and provided important guidance for our research. We have studied the comments carefully and have made corrections that we hope will meet with your approval. The main corrections in the paper and the responses to the reviewer's comments are as follows:

The authors revealed the effects of N fertilizer reduction on soil ecosystem services under straw-return conditions from exogenous inputs and microbial perspectives, as well as potential microbial relationships. The work is rewarding, but I have some suggestions that need attention so that I can improve the quality of the manuscript.

1. In my opinion, the N+PK treatment in this work is a regular fertilization practice in the field, so when describing the results, the main description of the results should be "decreasing with the application of N fertilizer" rather than "increasing", and the description of the results is likely to cause confusion to the readers.

**Reply:** Thank you for the comment. We have revised these sentences.

Regarding greenhouse gas emissions, with decreasing N fertilizer application levels, $CO_2$ and $N_2O$ emissions gradually decreased. (Lines 308-309)

2. Lines 83-106, try to reduce this section, too long a description leads to less readability.

**Reply:** Thank you for the comment. We have reduced this section.

3. Lines 138-141, I think the author has made a writing error here. The text describes 4 treatments instead of 5.

**Reply:** Thank you for the comment. We have revised the text.

A randomized complete block design consisting of 4 treatments with 3 replications was adopted. (Lines 125-126)

4. Lines 187-213, this section needs to be supplemented with the necessary references.

**Reply:** Thank you for the comment. We have added the necessary references.

5. Lines 551-563, I think this section is a description of Fig. 6, however there is no discussion of specific microbial species within module communities, please add content.

**Reply:** Thank you for the comment. We have added relevant content.

The *Lasiosphaeriaceae*-driven module 1 and *Terrimonas*-driven module 3 communities may be involved in maintaining soil ecosystem multifunctionality. (Lines 566-567)

This is an interesting manuscript that could be accepted and published, after revising the above issues.

Thank you for your valuable comments. We hope our responses will meet with your approval.

---

## Author Response (AR2)

List of responses

Dear Editor

Thank you for your comments concerning our manuscript entitled "**Moderate N fertilizer reduction with straw return modulates cropland functions and microbial traits in a meadow soil**" (Manuscript ID No. egusphere-2023-2498). Those comments are all valuable and hopeful for revising and improving our manuscript, as well as the important guiding significance to our research. We have studied comments carefully and have made the correction which we hope meet the approval. The main corrections in the paper and the responses to the editor comments are as following:

1. L42 & 43 - causality not appropriate, these are correlations

**Reply:** Thanks for your comments. The sentences have been revised.

The *Lasiosphaeriaceae* within module 1 community showed significant positive correlations with straw degradation rate, C and N release, while the *Terrimonas* within module 3 community showed a significant positive correlation with production, which were conducive to soil multifunctionality. (Lines 41-44)

2. L99 - introduction of the word 'traits' here requires support before - for instance I suggest inserting the word as an equivalent to the properties and function, in the paragraph (L85-95) discussing those traits.

**Reply:** Thanks for your comments. The sentences have been revised.

Microorganisms contribute to soil functions by modulating microbial traits (e.g., function, community composition and succession), which are influenced by different N fertilizer input levels (Bradford et al., 2014; Chen et al., 2019a). (Lines 87-89)

3. L100 - Similarly, the uncommon term 'ecosystem disservices' should be introduced earlier (e.g., L69)

**Reply:** Thanks for your comments. The necessary content have been added.

However, crop fields suffering from superabundant exogenous materials may result in ecosystem negative effects. For example, excess organic materials usually have low reutilization efficiency

(Hou et al., 2020); the majority of N in straw is released into the atmosphere as oxynitride, such as nitrous oxide ($N_2O$) (Wang et al., 2019; Sun et al., 2021); Subsequent literature highlighted that straw return significantly elevates greenhouse gas emissions so that less than 15% of straw-derived N can be transformed into soil and become SOM (Yin et al., 2018; Wu et al., 2019). Moreover, as you suggested later, we have replaced ecosystem disservices by ecosystem negative effects. (Lines 68-73)

4. L114-117 - rename the treatments OR clarify that the only the urea fertilizer is being referred to with the titles '0.75N, 0.5N, or 0N', because these titles do not account for the N in DAP that was added to all plots (DAP is 18% N, or 45 kg N h-1 y-1)

**Reply:** Thanks for your comments. Due to my negligence, the original text incorrectly described the addition form of phosphate fertilizer. In this study, calcium triple superphosphate is the only phosphorus source in the soil rather than diammonium phosphate. The necessary explanation have been added and the incorrect description has been corrected.

Four treatments received different N fertilizer input levels after straw return to the field for 4 years as follows: (1) regular chemical fertilization, N+PK (300 kg urea (N 46%) $ha^{-1}$ $yr^{-1}$, 250 kg calcium triple superphosphate ($P_2O_5$ 46%) $ha^{-1}$ $yr^{-1}$, 150 kg potassium chloride ($K_2O$ 50%) $ha^{-1}$ $yr^{-1}$); (2) 25% reduction of N fertilizer, 0.75N+PK (225 kg urea $ha^{-1}$ $yr^{-1}$, 250 kg calcium triple superphosphate $ha^{-1}$ $yr^{-1}$, 150 kg potassium chloride $ha^{-1}$ $yr^{-1}$); (3) 50% reduction of N fertilizer, 0.50N+PK (150 kg urea $ha^{-1}$ $yr^{-1}$, 250 kg calcium triple superphosphate $ha^{-1}$ $yr^{-1}$, 150 kg potassium chloride $ha^{-1}$ $yr^{-1}$); and (4) no N fertilizer, PK (250 kg calcium triple superphosphate $ha^{-1}$ $yr^{-1}$, 150 kg potassium chloride $ha^{-1}$ $yr^{-1}$). Urea is the only nitrogen source in the soil. (Lines 112-120)

5. L286 - please describe how the heatmaps were constructed (software, assumptions)

**Reply:** Thanks for your comments. The necessary content have been added.

To construct the relationship between fertilization, soil function, and microbial traits, two heatmaps were constructed in this study (Origin 2022). The first heatmap was constructed to reveal the associations between cropland properties with microbial module comminuties. And another heatmap was constructed to reveal the associations between microbial traits and fertilizers, soil properties, greenhouse emissions and ecosystem multifunctionality. (Lines 288-292)

6. L295 - replace 'ecosystem services' with appropriate term (e.g., cropland properties, cropland traits), because these are not ecosystem services (ES). ES are services that are supporting (water infiltration, nutrient cycling, C sequestration), regulatory, provisioning, or cultural, provided by the environment, that directly or indirectly support humans. The functions described in this manuscript contribute to supporting ES, but are not in themselves ES. Also, that is a different topic than multifunctionality. Throwing around these terms will frutrate readers.

**Reply:** Thanks for your comments. In the original paper, we took soil fertility, microbial function, crop yield, etc., as indicators of ecosystem services, and greenhouse gas emissions as indicators of ecological dis-services. As you suggest, there are limitations to expressing these properties in terms of ecosystem services. After careful consideration, we have decided to replace all descriptions of "ecosystem services" with "cropland properties", "cropland traits" and "soil functions". Moreover, "ecosystem dis-services" was replaced by "ecosystem negative effects".

The relevant sentences have been revised.

7. L309 - Fig 1k, 1l (not 1m)

**Reply:** Thanks for your comments. The sentence has been revised.

For greenhouse gas emissions, with the decrease in N fertilizer application levels, $CO_2$ and $N_2O$ emissions gradually decreased (Fig. 1k, l). (Lines 311-312)

8. L310 - Fig 1m (not 1l)

**Reply:** Thanks for your comments. The sentence has been revised.

No significant difference was observed in $CH_4$ emissions under the different fertilization treatments (Fig. 1m). (Lines 312-313)

9. L311 - remove 'as expected' as this level of detail was not provided in the hypotheses (or change hypotheses)

**Reply:** Thanks for your comments. We have deleted 'as expected'.

The 0.75N+PK treatment achieved the highest multifunctionality index (0.61), followed by N+PK (0.32), 0.5N+PK (-0.34) and PK (-0.59) (Fig. 1p). (Lines 314-315)

10. Fig 1n: please change 'straw biomass' to 'AG biomass' to be consistent. Easy to confuse with the straw litter bag assay.

**Reply:** Thanks for your comments. We have revised this figure.

And you can see these in Fig.1.

11. Fig 2d: replace 'mineralization' with 'emissions' or 'flux rate' (also l.411)

**Reply:** Thanks for your comments. We have revised these figures.

And you can see these in Fig.1 and Fig.2.

12. L395 - the text here and in the fig 4 caption should say '…close correlations of N input (fertilizer and straw return)…'

**Reply:** Thanks for your comments. The sentence has been revised.

The heatmap showed the close correlations of N input (fertilizer and straw return) with soil stoichiometry and microbial traits (Fig. 4). (Lines 396-397)

13. L402 &L404 - replace 'affecting' with 'correlated to'

**Reply:** Thanks for your comments. The sentence has been revised.

The results indicated that the N input level, straw biomass and soil C:N ratio were the most prominent abiotic factors correlated to the ecosystem multifunctionality index, while some biotic factors, such as the abundance of genes encoding cellulose-degrading enzymes, significantly correlated to the ecosystem multifunctionality index. (Lines 402-405)

14. L424-426 - why aren't these OTUs assigned taxonomy here?

**Reply:** Thanks for your comments. We have added the relevant content.

Bacterial *Terrimonas* (in module 1), Myxococcales (in module 2) and *Terrimonas* (in module 3) were highlighted as essential predictors of soil ecosystem multifunctionality, and fungal Lasiosphaeriaceae (module 3) was also found to be an important variable for predicting its changes. (Lines 426-408)

15. L505 - fungal C:N is typically less than 20, for instance 15.7:1 (250:16:1 for C:N:P) with a wide range (https://www.frontiersin.org/journals/microbiology/articles/10.3389/fmicb.2017.01281/full).

Perhaps you could provide a reference to support the value of 20:1?

**Reply:** Thanks for your comments. We have revised these inappropriate descriptions.

Previous studies have shown that C:N:P ratio of soil microbial biomass was stable at 60:7:1 (Cleveland et al., 2007) and fungal biomass C:N ratio was higher than this ratio (nearly 15.7:1) (Zhang et al., 2017). Generally, Bahram et al. (2018) concluded that higher C:N ratio may promote fungal abundance and decrease bacteria:fungi ratio. (Lines 505-508)

**Reference:**

Bahram, M., Hildebrand, F., Forslund, S.K., Jennifer L. A., Nadejda, A. S., Nadejda, A. S., Johan, B., Sten, A., Luis, P. C., Helery. H., Jaime. H., Marnix, H. M., Mia, R. M., Sunil, M., Pål, A. O., Mari, P., Sergei, P., Shinichi, S., Martin, R., Leho, T., Peer, B.: Structure and function of the global topsoil microbiome, Nature 560, 233 – 237, doi: https://doi.org/10.1038/s41586-018-0386-6, 2018.

Cleveland, C.C., Liptzin, D.: C:N:P stoichiometry in soil: is there a "Redfield ratio" for the microbial biomass?, Biogeochemistry 85, 235–252, doi: https://doi.org/10.1007/s10533-007-9132-0, 2007.

Zhang, J., Elser, J.J.: Carbon:Nitrogen:Phosphorus Stoichiometry in Fungi: A Meta-Analysis. Front. Microbiol. 8:1281. doi: https://doi.org/10.3389/fmicb.2017.01281, 2017.

16. L524, 525, 537. Please replace the term 'ecosystem services'. These are ecosystem or soil functions, but not ecosystem services.

**Reply:** Thanks for your comments. We have revised these sentences.

As we described earlier, ecosystem services have been replaced in full.

It is therefore necessary to further explore the potential associations between microbial traits and soil functions under diverse N fertilizer input levels. (Lines 525-527)

Numerous studies have shown that core microbiota play a vital role in maintaining the stability of soil microbial function and the complexity of microbial networks and then promoting soil nutrient cycling and other soil functions; (Lines 540-542)

In addition, we have also replace 'ecosystem services' with appropriate term in the whole manuscript.

17. L540 & L550 - link to the OTU number used as a label in the results, to be consistent (i.e., see remark for L 424-426).

**Reply:** Thanks for your comments. We have revised these sentences before. The OTU number and taxonomy are consistent.

Specific thanks to you for your insightful comments

We appreciated for Editor's warm work earnestly and hope that the correction will meet with approval.